# Role of Health Professionals Regarding the Impact of Climate Change on Health—An Exploratory Review

**DOI:** 10.3390/ijerph18063222

**Published:** 2021-03-20

**Authors:** Julien Dupraz, Bernard Burnand

**Affiliations:** Center for Primary Care and Public Health (Unisanté), University of Lausanne, 1010 Lausanne, Switzerland; bernard.burnand@unisante.ch

**Keywords:** climate change, health professionals, role, intervention

## Abstract

Health professionals are increasingly urged to act to protect individuals and populations against the negative effects of climate and environment change on health. However, the amount of evidence supporting initiatives to that end is unknown. We explored the literature examining the awareness, preparedness, and role of healthcare professionals to inform about the impact of climate change on health on the one hand, and literature about the effectiveness of interventions mediated by health professionals aiming at reducing the environmental impact of human activities on the other hand. We included 137 articles published between 2000 and 2020, mostly in general medical and nursing journals. The typical article was a perspective, commentary, or other special article aimed at alerting readers about the impact of climate and environment change on health. We identified 22 studies, of which only two reported interventions. Despite increasing efforts of health professionals to address climate and environment change and related health risks, health literature supporting such efforts remains scarce, and studies assessing the effectiveness of interventions are lacking. We need appropriate evidence to indicate which interventions should be prioritized, considering that the association of health issues with climate and environment change could constitute an effective lever for change.

## 1. Introduction

The negative effects of climate and environment change on human health have been reported for several decades [1]. Climate change already produces negative effects on human health, worldwide, and will have additional harmful effects in the future. The World Health Organization emphasizes the risks for human health due to climate changes in a “health topic” [2]. The “Commission on Health and Climate Change” established in 2015 by The Lancet reports every year on the situation, presenting various data and indicators on the effects of climate change on the health of populations and on the political and health policy decisions aimed at mitigating the harmful effects of climate change on health. The most recent updated report was published in December 2020 [3]. More frequent extreme temperatures (e.g., heat waves) may endanger human life and increase morbidity (e.g., through dehydration, cardiovascular events, infectious conditions), leading to higher health services use and losses in working hours. The expansion of vector- and water-borne infections is occurring in many regions. In addition, because of the effects of climate change on soil, agriculture yield is decreasing, contributing thus to limit access to appropriate food, especially in already deprived areas. Moreover, healthcare is contributing to carbon dioxide emissions, corresponding to an average close to 5% worldwide [3] and higher in several developed countries. For several years, different initiatives have been targeting healthcare and public health professionals, inciting them to contribute actively to protect individuals and populations against the negative effects of climate change on health. For example, the World Organization of Family Doctors (WONCA) produced a “Declaration calling for family doctors of the world to act on planetary health” [4], and the Canadian Association of Physicians for the Environment published in 2019 a toolkit for physicians to fulfil this goal [5]. These initiatives are welcomed; however, their recommendations do not appear to be supported by much evidence. Healthcare professionals are usually trusted by the public [6], it is therefore crucial that the recommendations and actions proposed by healthcare professionals, collectively or individually, are based on robust and direct evidence.

Therefore, we need to know which information and evidence is available in the literature to better design possible interventions involving healthcare professionals, including research and development programs, to contribute to tackle the harmful effects of climate and environment change on the health of individuals and populations. Thus, the aim of this exploratory review was to provide an overview of existing scientific articles about the awareness, preparedness, and role of healthcare professionals to inform individuals, communities, and populations concerning the relationships among the environment, climate change, and health. To fulfill our ultimate objective, we searched for the existence and effectiveness of interventions mediated by healthcare professionals to motivate individuals and communities to adapt lifestyles reducing the impact of human activities on climate and environment change.

## 2. Materials and Methods

We scoped a large range of article types and made simple descriptive analyses to illustrate our findings, which we reported narratively. According to our exploratory intent, we used one database only, and one reviewer performed data extraction. We included articles addressing the awareness, preparedness, and possible roles of healthcare professionals regarding climate and environment change, as well as interventions of healthcare professionals intended to contribute to mitigate these changes. We included articles published between 2000 and 2020 in the health literature, reporting explorations, surveys, and interventions targeting the aforementioned aims. We included original studies, quantitative and qualitative, or mixed, reviews, and various types of reports. We thus purposefully searched a large range of published articles. We excluded articles written in a language not mastered by the authors and articles for which we could not obtain a full-text version.

An initial pilot search was conducted on the PubMed, Embase, and Epistemonikos databases that indicated that searching on PubMed only was sufficient to fill our exploration purpose. The most recent search update was conducted on 23 October 2020. The search strategy included the following components: target population (healthcare professionals), context (climate and environment, and health issues), information (knowledge implementation, guidance, training, intervention). The actual search strategies were constructed and tested with the support of a senior librarian (Appendix A presents the search strategy). The initial selection was based on titles with the aim of excluding articles that were obviously out of scope. In case of uncertainty, the abstract was examined. This step was achieved by one reviewer (J.D. or B.B.). The initial selection was examined by the second reviewer and the final list of articles to be considered for full-text analysis and potential inclusion was obtained by consensus. Similarly, the list of included articles was confirmed by consensus.

We extracted the following descriptive information from the included articles: publication year, author number, journal type (general, specialized), country of realization, page and reference count, type of article (editorial, letter, special article (e.g., perspective, commentary, debate, report), position paper, narrative review (no method indicated), study (formal review, intervention, survey, qualitative method)). We also intended to record the aim of the article (explicit or implicit) and the propositions or recommendations made. One reviewer performed the extraction (J.D. or B.B.); a double extraction was performed on a series of 10 articles, which indicated an excellent agreement (close to full agreement), with the exception of the recommendations and propositions of articles’ authors that we did not consider reliable enough to perform a quantitative analysis. We performed a simple descriptive analysis of the information retrieved from all included articles, which we reported in a table form. We undertook a more detailed analysis of the subgroup of studies. Within the analysis, we described the studies’ aim, method, and target participants and participation proportion, to which we added a narrative comment. We did not intend to perform a qualitative analysis of a very heterogeneous and large group of editorials, commentaries, and perspective or opinion articles in this initial approach. In addition, we did not formally assess the quality of the included studies given our exploratory purpose, the various designs used, and the large diversity in the reported methods and results.

## 3. Results

### 3.1. Global Analysis

The literature search allowed us to include 137 articles published in 84 journals (full list is available in Appendix A). Twenty-three journals published more than one article (up to seven in our selection). The article selection flowchart is presented in Figure 1. The characteristics of the included articles are presented in Table 1. Most articles were published in general medical and nursing journals. About half of the retrieved articles were published between 2016 and 2020 and were written by authors based in the USA, whereas much fewer articles were published by European authors. These articles were often relatively short with a median of four pages and 16 references. Forty-two percent of these articles were single authored, and the median number of authors was two. The typical article was a perspective, commentary, or other special article aimed at informing and alerting readers about the links between climate and environment change and human health.

Most articles (98, 72%) did not explicitly present their objective. However, proposals and recommendations for the possible roles of healthcare professionals were frequently made. Most often, they concerned information and advocacy targeting the public and decision-makers, proposals of actions within healthcare organizations or practices, and enhanced education and training of healthcare professionals. Less often, direct information and advocacy targeting patients and changes in personal actions (role model) were indicated.

### 3.2. Analysis of Actual Studies

Twenty-two articles presented a study (three literature reviews, 12 surveys, five qualitative and mixed methods studies, and two interventions studies). They are described in Table 2, classified by study type.

#### 3.2.1. Reviews

The first review indicated the increased vulnerability of elderly individuals to extreme heat and the role of general practitioners in identifying them and to implement strategies to minimize their risks [7]. The second review was conducted to support the recommendations of the American College of Physicians to call physicians to act to “improve human health and avert dire environmental outcomes” [8]. The third review identified the climate change and sustainability topics to be included in nursing education [9].

#### 3.2.2. Surveys

Most identified surveys targeted healthcare professionals. Three surveys targeted public health departments’ officers [10,11,12] indicating that many of them acknowledged that climate change is a relevant threat for public health, but very often they lacked information, expertise, or resources to address that threat. In two related surveys, a majority of national and international American Thoracic Society respondent members indicated that climate change is happening, already affecting patients’ health [13,14]. Five surveys targeted nursing and medical education curricula, which indicated perceived needs to include climate change and environmental sustainability topics in their programs [15,16,17,18,19]. One, limited, survey among patients visiting a general practice in Israel [20] and a larger U.S. population survey [21] indicated that, although aware that global warming may harm health, patients and population need, and may expect, more complete and specific information about health in relation with climate and environmental changes. Three surveys had important limitations either due to a small size or the methods used, and three other surveys had response rates below 20%.

#### 3.2.3. Qualitative and Mixed Methods Studies

The qualitative and mixed methods studies also indicated needs for specific professional training on climate and environment change and health and the necessity to support messages and actions targeting patients and the public [22,23,24,25]. A case study of Australian health agencies indicated that available competencies and frameworks were transferable to action on climate change, such as health promotion practices oriented toward active and sustainable transport and healthy and sustainable food supply, for instance [26].

#### 3.2.4. Intervention Studies

One training and education intervention targeted hospital workers (housekeeping and food departments) and indicated improvements in the use of cleaning chemicals and waste disposal [27]. A pilot study indicated the feasibility and high adherence to an adult education program about energy use and sustainability, combined with mindfulness meditation [28].

## 4. Discussion

In this exploratory review, we identified more than 100 published articles addressing the awareness, preparedness, and role of healthcare professionals in the context of climate and environment change. We included a large and heterogeneous group of articles. The majority of them were commentaries, editorials, and opinion or perspective texts. We found 22 actual studies presenting quantitative or qualitative results. Our ultimate goal was to find evaluations of interventions aimed at modifying the decisions, behaviors, or habits of patients, healthcare professionals, or other stakeholders to mitigate climate change consequences. We found only two such studies. Almost all articles intended to alert on the increasing risks of climate and environment change on humans, human health, and the planet, contributing thus to increase the awareness of healthcare professionals. However, this number of 137 publications must be weighed against the very large number of existing health-related journals. The number of readers who eventually read these articles may be limited. However, various entities as well as international and national associations of healthcare professionals provide information, position statements, recommendations, or toolboxes on websites. Table 3 presents several examples of those, as we did not aim to provide a systematic description and analysis of their contents.

In order to compare our results with those of similar reviews, we found the protocol of a scoping review with a comparable aim (“mapping the range and nature of existing evidence regarding health professionals’ knowledge, attitudes, perceptions, and practices regarding climate change and health impacts and the challenges they face”) [29]. Two other published protocols of scoping reviews focused on hospital climate actions and assessment tools [30], on the one hand, and on teaching the relationship between health and climate change [31], on the other hand. However, we did not find the corresponding reviews in the literature. Another review had a related but different purpose, which focused on the environmental competencies required and pedagogic approaches used to include sustainability in curricula in nursing education [9]. In addition, a review focused on nurse preparedness to face the growing number of natural disasters, which concluded that although increasing, nurse preparedness should be further improved [32]. The 2016 recommendations developed by the American College of Physicians in a position paper on climate change and health [8] were based on a literature review of studies and reports about the association of climate change and human health.

Various initiatives and propositions about the possible roles of healthcare professionals were presented in the retrieved articles. Several proposed actions and changes aimed at increasing the pre-, postgraduate, and continuing education of healthcare professionals about climate and environment change, their effects, and their already-occurring and future consequences on human health. These actions are supposed to improve the awareness and preparedness of healthcare professionals. However, we found only two reports of studies aimed at assessing the effectiveness of interventions to motivate individuals, patients, and communities to intentionally reduce the impact of human activities on climate and environment change through lifestyle changes. The co-benefits of lifestyle changes on both human health and the climate should be better supported. Eating less meat products and, when distance permits, walking or biking instead of using transportation that contributes to global warming and environmental pollution; these constitute two examples of individual and collective actions that should be more intensively promoted by healthcare professionals. Nevertheless, we need appropriate evidence of the effectiveness of such interventions and corresponding indicators to allow their development, support, and improvement. Indeed, the various recommendations retrieved in this review were chiefly based on logical reasoning, given the lack of evidence of effective interventions.

Collective awareness of the association of climate and environment change with human health has been relatively limited until recently. Indeed, a new indicator introduced in the 2019 report of The Lancet Countdown on health and climate change indicated little connectivity between the clusters “health” and “climate change” in searches conducted on Wikipedia [33]. However, the 2020 report indicated a 24% increase in information seeking about health and climate change from 2018 to 2019, which was mainly driven by the initial interest of individuals in health [3]. The 2020 report also emphasized a “growing response from health professionals” (e.g., decarbonization efforts, linkage with meteorological services for health services planning). In addition, the coverage of health and climate change in the media, worldwide, almost doubled from 2018 to 2019.

Most included articles present narrative descriptions of the relations among climate and environment change and health, opinions, position statements (e.g., for healthcare sector decarbonization), mitigation strategies (e.g., to reduce the impact of heatwaves on elderly individuals), or recommendations to reach global or specific sustainability targets supposed to limit the negative effects of climate and environment change on health and the planet. Actually, the content of many of these special articles or editorials is not very different from the perspective paper published by McCally and Kassel in 1990 [34].

Our exploratory review has several limitations, already indicated in the methods section. The literature search was not exhaustive, despite the tested search equation. We conducted our literature search in one database only. We excluded several publications not written in English, but they did not report studies. In addition, although it fitted our purpose, the search scope might have been too broad. Indeed, it could have been more fruitful to focus on more specific questions or interventions. Moreover, we provided a limited description and analysis of the publications and studies. The 12 published surveys indicate that a majority of healthcare professionals believe that climate and environmental changes are real and constitute a threat for human health, and that healthcare professionals have the duty to alert and act. However, given the relatively limited response rates to these surveys, these results may be biased. Finally, despite the limitations of this exploratory review, we believe that the provided depiction of the literature offers a rather reliable image.

## 5. Conclusions

Our exploratory review indicated that the role of healthcare professionals, individually or collectively, through professional organizations, could include informing and alerting patients, individuals, communities, and decision-makers about the association of climate and environment change with health and the need to act to limit and mitigate these risks to protect health. Various channels have been used such as websites of healthcare organizations, interventions targeting decisions-makers, and highly visible reports such as The Lancet Countdown on health and climate change edited anew each year. However, there are relatively few publications in the health literature, and we lack studies assessing the effectiveness of various interventions to be implemented by healthcare professionals, directly (e.g., promoting lifestyle changes at the individual or community level) or indirectly (e.g., promoting structural changes through laws and regulations, as well as organizational changes in hospitals and health systems). Nevertheless, we believe that healthcare professionals have a professional public health duty to foster the assessment and implementation of effective interventions, to improve the education of their peers, and to keep informing and alerting various audiences through potentially appropriate communication interventions [35,36]. Indeed, the association of health issues with climate and environment change could constitute an effective lever for change.

## Figures and Tables

**Figure 1 ijerph-18-03222-f001:**
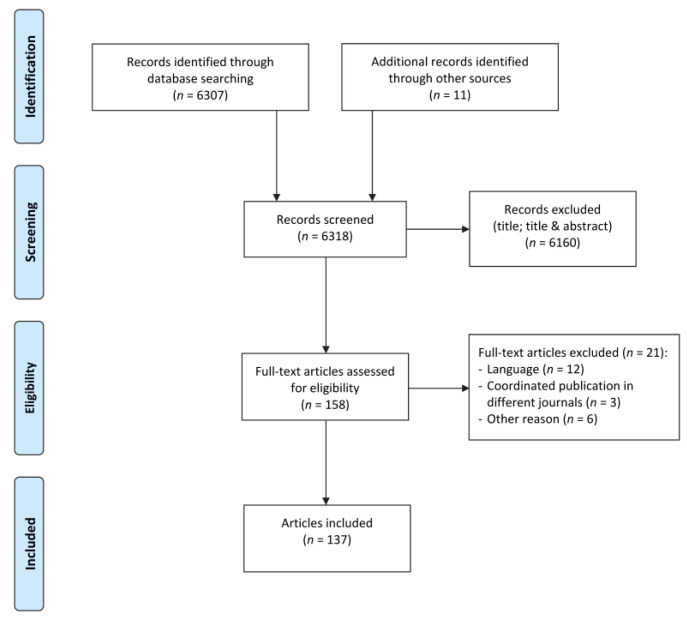
Flow diagram from identified to included reports or studies.

**Table 1 ijerph-18-03222-t001:** Description of the typology of 137 included articles.

**Journal Category**	***n* (%)**
Nursing and midwifery	40 (29)
General medical journal	36 (26)
Public health	21 (15)
Specialty medicine	15 (11)
Family medicine	13 (9)
Other	12 (9)
**Article Type**	***n* (%)**
Special article (perspective, commentary, debate, report, etc.)	78 (57)
Editorial	25 (18)
Study	19 (14)
Letter	6 (4)
Narrative review (informal)	6 (4)
Position paper (professional association)	3 (2)
**Number of Authors Median**	**Number of Authors Mean**
2 (p25: 1–p75: 3)	2.6
**Number of Pages Median**	**Number of Pages Mean**
4 (p25: 2–p75: 8)	5.1
**Number of References Median**	**Number of References Mean**
16 (p25: 6–p75: 33)	24.3
**Country of Authors**	***n* (%)**
USA	68 (50)
Australia and New Zealand	28 (20)
Europe and UK	20 (15)
Other countries	11 (8)
International (authors of >1 country)	10 (8)
**Publication Year**	***n* (%)**
2000–2005	6 (4)
2006–2010	34 (25)
2011–2015	31 (23)
2016–2020	66 (48)

**Table 2 ijerph-18-03222-t002:** Description of 22 studies; reports of elements from published documents *.

Study	Aim	Method	Participants	Comment
Literature review (*n* = 3)				
Wilson, 2011 [7]	Review evidence on heat-related health risk and the role of general practitioners in reducing heatwave-related morbidity in older people	Formal literature review with focus on heatwaves and health in elders and on related role of professionals	Search in 3 databases; 43 articles selected from 661 identified, narrative report of findings	Multifactorial increased vulnerability of elders to extreme heat. GPs play a crucial role in identifying those at risk and implementing strategies to minimize the risks of mortality and morbidity
Crowley, 2016 [8] †	Review to support the position paper of the American College of Physicians on climate and health	Collective work of committee, no details of methods beyond brief description of search	Review of available studies (2 databases), reports, surveys, policy documents, websites, and other sources; 144 references	Narrative report of review to support 5 recommendations proposing a series of individual and collective active roles of physicians to alert, inform, and act to “improve human health and avert dire environmental outcomes”
Lopez-Medina, 2019 [9]	Review environmental competencies required and pedagogic approaches used to embed sustainability in nursing education curricula	Formal review of publications about climate change and sustainability topics that nurses need to know and have skills and competencies in, and related pedagogical approaches; thematic analysis	Search in 28 databases, 4 languages, 2004–2017; 2 articles selected from 620 abstracts out of 4718 titles	Topics such as use of resources, food, health promotion, globalism, disease management, and environmental impact of delivering healthcare, should be included in nursing education to support the nursing profession’s response to the often-vulnerable people they care for
Survey (*n* = 12)				
Maibach, 2008 [10]	Understand how directors of local public health departments view and are responding to climate change as a public health issue	Telephone survey, randomly selected, stratified sample, 4 primary questions, 52 items	Local health department directors, USA; *n* = 217, 133 responses (61%)	Majority of respondents perceived climate change to be a growing problem but lacked knowledge about climate change. Small minority had made climate change adaptations a top priority for their health department. Majority needed additional support and funding
Bedsworth, 2009 [11]	Gage concerns about public health impacts of climate change, programs in place to mitigate health effects, information, and resource needs	Development of questionnaire with experts. Initial letter, followed by e-mails: web-based survey and periodic reminders. 4 main questions	Local public health officers, California, USA; *n* = 61, 34 responses (56%)	Most felt that climate change poses a serious threat to public health and that they lack resources or information to cope with that threat. Nonetheless, implementation of mitigation programs was ongoing
Carr, 2012 [12]	Assess local health department officials’ perceptions and preparedness related to climate-sensitive health areas	Online survey adapted from Maibach 2008, 4 main questions, 3 reminders	New York State County health departments, USA; *n* = 56, 30 responses (54%)	39% respondents perceived climate change as a relevant threat, 3/4 felt they were knowledgeable about the potential health effects of climate change. Lack of specific expertise to address climate-related issues. Several relevant programs already in place. Large uncertainty about needs of additional resources
Sarfaty, 2015 [13]	Assess perceptions of clinical experiences with, and preferred policy responses to, climate change of American Thoracic Society (ATS) members	Web-based survey e-mailed by ATS President, up to four reminders. Validated questionnaire (46 items) + open questions	ATS U.S. randomly selected members; *n* = 5500, 915 responses (17%)	Most respondents felt that climate change is happening (89%), driven by human activity (68%), relevant to patient care (65%), with effects already present among patients (increase in chronic disease severity, allergic symptoms, severe weather injuries); support for educating the patients, public, and policy-makers on human health effects of climate change
Sarfaty, 2016 [14]	Assess perceptions of clinical experiences with, and preferred policy responses to, climate change of American Thoracic Society (ATS) members	Web-based survey e-mailed by ATS President, up to four reminders. Validated questionnaire (46 items, adapted from Sarfaty 2015) + open questions	ATS international members; *n* = 5013, 489 responses (10%; 68 countries)	Large majority of respondents indicated that climate change is happening (96%), driven by human activity (70%), relevant to patient care (80%), with effects already present among patients (increase in chronic disease severity, allergic symptoms, severe weather injuries); support for educating the patients, public, and policy-makers on human health effects of climate change
Kirk, 2002 [15] †	Ascertain views of senior nurse academics on global environment and nurse education	Pilot study. Convenience group. Brief e-mailed questionnaire (1 reminder), agreement with 13 statements (very succinct description)	Senior academics of UK nursing schools; *n* = 68, 18 responses (31%)	Very brief presentation of survey results in a general article. Majority of respondents agreed that awareness of global environment issues is important, and lack of awareness undermines the ability of nurses to contribute to debate and decision-making
Teherani, 2017 [16]	Determine which and when a set of “sustainable healthcare education” (SHE) objectives should be included in the medical education continuum	Modified Delphi approach to conduct a two-step survey of SHE experts; descriptive statistics and item-level content validity index (CVI)	*n* = 82 experts, 52 responses (63%; physicians, academics); 15/21 SHE objectives with CVI > 78%	13 objectives for preclinical years and 6 for clinical years to prepare physicians to care for patients who experience the impact of climate and environment on health and advocate for sustainability of the health systems in which they work
Cruz, 2018 [17]	Assess attitudes of nursing students toward climate change and environmental sustainability and their inclusion in the nursing curricula	Environmental Sustainability Attitudes in Nursing Survey 2 (SANS-2, self-administered questionnaire); 5 questions about climate change and environmental sustainability	Convenience sample, nursing students in Egypt, Iraq, Palestinian Territories, and Saudi Arabia; 1059 students responded (participation rate not mentioned)	55–60% of students mildly, somewhat, or strongly agreed with the 5 statements; less than 20% disagreed; authors propose to include climate change and environmental sustainability in nursing curricula
Eide, 2019 [18]	Enquire whether nursing schools’ students receive education about climate change and environmental sustainability	Survey using a 17-item questionnaire	*n* = 213 U.S. nursing schools, 81 responses (38%)	Respondents indicated existing education about environmental issues (18%), sustainability (26%), and climate change (2%); authors propose to introduce education about these 3 themes to improve nurses’ ability to respond to the predicted clinical and public health changes
Kemper, 2020 [19] †	Understand whether it is feasible to survey pediatricians regarding advocacy about climate change	Pilot survey; selected group of pediatricians: members of the Academic Pediatric Association having a special interest in environmental health	*n* = 83, 66 responses (80%); (very brief description in a more general article)	Most respondents felt responsible to understand the impact of climate change on human health; about half received formal education about health effects of climate change. Many of this selected group were already engaged in climate advocacy
Guggenheim, 2016 [20]	Examine whether visiting a medical practice can promote reflection on well-being and health related to environmental issues	Limited survey, paper forms in waiting room of general practice (4 GPs); questions on environment issues followed by open questions (ways to change personal habits)	Patients of a group practice in Israel; *n* = 107 participants	Very brief presentation of survey results. Participants indicated significant interest in and concern for environmental issues and willingness to contribute to their improvement
Maibach, 2015 [21]	Describe population awareness of health effects of global warming, support for action, trust in information sources	Online survey, closed and open-ended questions (average completion time: 29 min)	Representative sample of US adults; *n* = 1275 responses (recruitment rate 14%, completion rate of 57%)	Most respondents reported that global warming can harm health (61%), but <1/3 identified types of harm or who is most likely to be affected. Moderate support for expanded public health response
Qualitative and mixed methods study (*n* = 5)				
Sheffield, 2014 [22]	Identify healthcare providers’ perceived health threats of climate change, role as informers and detectors of disease for their patients	Qualitative: focus groups (records, transcription, grounded theory method, and axial coding)	28 healthcare providers, 5 focus groups; hospitals, clinics, home health services in a low-income community, New York City, USA	Healthcare providers were interested and receptive to climate change and public health, engaged in discussing environmental problems and local health impacts, seeing their role in adaptation to climate change, supportive of clear public health messages
Valois, 2016 [23]	Identify family physicians’ educational needs about health impacts of climate change and beliefs and conditions related to participating in continuing medical education (CME)	Mixed method; qualitative interview and quantitative online questionnaire	23 family physicians involved in CME (convenience sample) participated in interviews and 14 filled a questionnaire; 12 sanitary regions in Quebec, Canada	Need for improved medical education on climate change and health; a 3-hour electronic CME training would be useful to support discussions and messages to patients
Singleton, 2018 [24]	Explore pharmacists and pharmacy technicians’ knowledge and understanding of the impact of pharmaceuticals on the environment and handling of pharmaceutical waste	Mixed methods. Purposive sampling of pharmacists and pharmacy technicians from all hierarchical levels in public, private urban and rural hospitals. Semi-structured face-to-face interviews and 5 yes/no questions	64 hospitals pharmacists and pharmacy technicians in 5 public and private hospitals (Queensland, Australia)	Lack of environmental knowledge regarding the impact of pharmaceuticals on the environment and lack of understanding of systems thinking concepts. Need for information and education
Völker, 2018 [25]	Investigate the determinants of whether physicians assess environmental history of their patients and provide environmental health advice	Mixed-methods: self-administered, structured questionnaire sent to a convenience sample of physicians of 3 selected hospitals (private/public, urban/suburban); and qualitative interviews with key informants	73 of 210 (35%) physicians replied to the survey; 6 key informants from health and environment sectors; Bangkok metropolitan area, Thailand	Limited knowledge and training of physicians in health-related environmental issues and rare routine discussion of eco-health links with patients. Need for revised education of physicians and coordination with healthcare system and government
Patrick, 2011 [26]	Investigate examples of health promotion practices addressing climate change and sustainability issues within healthcare settings	Case studies of healthcare agencies; 6 semi-structured individual interviews and group interviews to cross-check individual accounts and specific aspects of practice; analysis of agencies’ documents	5 Australian healthcare agencies that explicitly identified climate change as a priority; 5 health promotion practitioners and 5 community health practitioners or leaders with health promotion activities	Competencies and frameworks were transferable to action on climate change in these healthcare settings, such as health promotion practice oriented toward active and sustainable transport, healthy and sustainable food supply, mental health and community resilience, engaging vulnerable population groups, and organizational development
Intervention study (*n* = 2)				
Chenven, 2013 [27]	Improve knowledge, actions, and communication in front-line healthcare workers to contribute to better environmental results in hospitals	Training program to reduce waste, increase recycling, decrease use of cleaning chemicals (qualitative and quantitative results). No methods described, selected results presented (quotes, costs, and waste reduction in several hospitals), issued from reports and presentations	More than 2500 hospital workers (housekeeping and food departments); 11 employers in 4 U.S. regions	Development and implementation of a training and education model for front-line healthcare workers in hospital settings that supported systems change and built new roles for these workers. It empowered them to contribute to “triple bottom line outcomes” in support of “people” (patients, workers, community), “planet” (environmental sustainability, lower carbon footprint), and “profit” (cost savings for institutions)
Grabow, 2018 [28]	Pilot testing of an 8-week adult education program about energy use, climate change, and sustainability, combined with training in mindfulness meditation	Program developed by a department of family medicine and community health and a school of nursing. Pilot testing in community individuals. Assessment of participants’ household energy use, transportation, diet, health (depression, stress, health-related quality of life), and happiness. Focus group and interviews	16 individuals (aged 30–63), USA	Program well received by participants, high adherence rate; assessments feasible (high response rate); authors reported that the program was feasible and, if shown to be effective, could have the potential to reduce individual carbon footprints, while supporting personal health

* Often original quotes from published articles. † Study included as part of more global article.

**Table 3 ijerph-18-03222-t003:** Examples of websites of healthcare entities and associations calling for action on climate and environment change to protect health.

Entity—Association	Website	Examples of Specific Content
World Health Organization	www.who.int (accessed on 19 March 2021)	Health topic “Climate change”: -fact sheet -capacity building (toolkit)
The Global Climate and Health Alliance	www.climateandhealthalliance.org (accessed on 19 March 2021)	-events -initiatives
International One Health Coalition	www.onehealthplatform.com (accessed on 19 March 2021)	-media bulletin -events
World Medical Association	www.wma.net (accessed on 19 March 2021)	Resolution “Protecting the future generation’s right to live in a healthy environment”
World Organization of Family Doctors (WONCA)	www.globalfamilydoctor.com (accessed on 19 March 2021)	Working Party on the Environment: “Declaration Calling for Family Doctors of the World to Act on Planetary Health”
International Council of Nurses	www.icn.ch (accessed on 19 March 2021)	Position statement on climate and health
World Federation of Public Health Associations	www.wfpha.org (accessed on 19 March 2021)	Policy and advocacy: “Public health for the future of humanity”
American Medical Association	www.ama-assn.org (accessed on 19 March 2021)	Declaration “Climate change is a health emergency”
Australian Medical Association	www.ama.com.au/ (accessed on 19 March 2021)	
British Medical Association	www.bma.org.uk (accessed on 19 March 2021)	
Canadian Association of Physicians for the Environment	www.cape.ca (accessed on 19 March 2021)	“Climate Change Toolkit for Health Professionals”

## Data Availability

The data presented in this study are available on request from the corresponding author. The data are not publicly available due to legal and privacy issues.

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
