# Peer review of "Role of Health Professionals Regarding the Impact of Climate Change on Health—An Exploratory Review"

_ijerph, 2021, doi:10.3390/ijerph18063222_

Round 1

Reviewer 1 Report

I believe that this work could serve as a valuable survey of the literature in this important and clearly underserved domain. I sincerely hope that it is ultimately published (after revision) and prompts further work in this area. I do, however, have a few recommendations that I think could make the article appreciably stronger:

  • To me, explaining your methods is very important for a piece like this, as I want to know how you scoped your searches, processed what you found and so on. The phrase "The protocol of this exploratory review was neither published nor formally registered; it is however available from the authors." is strange to me. First of all, you do go on to introduce your methods at least somewhat in that section. Secondly, I would be as explicit as possible in doing so, within the paper. Again, I want to know how you went about identifying and processing your dataset (i.e., universe of articles you ultimately judged as relevant). For example, what was your "simple descriptive analysis"?
  • On a somewhat related note, since I assume it was part of your methods, I am confused about the transition from 137 articles to the 22 you focus on. What did the others do if not "present studies"? At the very least I would discuss why you seem to put them aside and transition to focusing on the 22.  At this point, they seem to suddenly disappear and we are left with only 22.
  • I would completely reorganize the results section so like-findings are clustered together in a clearer way. Right now, it seems to be a run-on paragraph of discrete findings with no clear logic to how they are ordered. Reorganizing might also allow you to pull out some higher-level headline results/findings that the others fall under.
  • And that results section is confusing as written - for example, you start with ’the review…’, ‘third review…’ and so on, and then start other sentences with things like “A majority of national and international American Thoracic…”. Aside from ordering, I think it needs some consistency in phrasing and generally greater clarity. 
  • I really like table 2. Some design and layout issues with things like awkward line breaks (and I would left rather than center-justify), but I think it provides a nice overview of your 22 papers.
  • In discussion, why use words like "few" rather than speaking specifically? Recommend saying "X articles did Y" rather than being vague.
  • You say “Only a few widely read journals are engaged to disseminate related messages (e.g. The Lancet).”  Did or could you look at the impact factors, citation counts, and/or other assessments of the resonance of the articles you include? Again, I think that kind of specificity would really strengthen your assertions.
  • Not sure why this is this in the discussion section: "We found the protocol of a scoping review with a similar aim («mapping the range and nature of existing evidence regarding health professionals’ knowledge, attitudes,  perceptions and practices regarding climate change and health impacts and the challenges they face») (29). Two other published protocols of scoping reviews focused on hospital climate actions and assessment tools (30), on the one hand, and on teaching the relationship between health and climate change (31), on the other hand " This seems more appropriate for the methods or perhaps findings section, but not discussion.

Reviewer 2 Report

see attached

Reviewer 3 Report

Remarks for the authors of the article:

Role of Health Professionals Regarding the Impact of Climate Change on Health. An Exploratory Review.The abstract presents the aim of the article. According to the authors, the aim is to draw attention to the impact of climate change and the environment on health. In the section Materials and Methods, the authors correctly stated the method of selection of the papers.The section Results correctly presents the flow diagram from identified to included reports or studies. In the section “Discussion”, the exploratory review was adequately examined and analysed. Technical side, the list of literature needs correction. However, the question arises whether the decides to adopt the paper aimed at presenting the exploratory review without more profound analysis. 

Round 2

Reviewer 1 Report

I feel that this article has substantially improved since the first review. I do have a couple of relatively minor remaining recommendations:

Methods: Some of my initial concerns persist. You seem to still sell the work short by saying "According to our exploratory intent, we purposely used a straightforward approach". I'm not sure what this adds - your approach was suitable for the work and this seems to suggest that you didn't really follow any method. You say later that "The protocol of this exploratory review was neither published nor formally registered; it is however available from the authors." However, isn't that what is in Supplementary Table 1? I am confused about what your protocol is other than your search and exclusion strategy, which you seem to include here and in the supplementary, although it might be more clearly (e.g., step-by-step) laid out. Last but not least, and I would defer to you and the editors around what is acceptable stylistically, but I wonder if you might just merge much of your search strategy (Supplementary Table 1) in with Figure 1?

Results: Just wanted to say that I really like this reorganization w/ subheadings. Makes it much clearer to me.

Table 2: I continue to really like this table, but also continue to think it needs some cleaning up. In various cases, words from one column appear to meld into words in the next. There must be some way to clean up (?). Could it be on pages laid out in landscape rather than portrait style (i.e., turned 90 degrees so the table is wider and shorter on each page)?

Reviewer 2 Report

I would like to commend the authors on addressing the various concerns raised during the first review.  The changes have made the paper for tighter and readable, especially through the use of subheadings. 

Below are some things for the authors to consider:

1: line 52... it is therefore crucial...

2. line 71 typo adressing, instead of addressing.

3. Line 81- could the search protocal be uploaded as supplementary material? 

4. line 108- there are quite a few descriptions. How about rewording?  eg we undertook a more detailed analysis of subgroup studies. Within the analysis, we describe...

5. Line 266 'already' seems redundant. 
